# A Compressed Sensing Measurement Matrix Construction Method Based on TDMA for Wireless Sensor Networks

**DOI:** 10.3390/e24040493

**Published:** 2022-03-31

**Authors:** Yan Yang, Haoqi Liu, Jing Hou

**Affiliations:** School of Electronic Information, Northwestern Polytechnical University, Xi’an 710000, China; yangyan7003@nwpu.edu.cn (Y.Y.); lhqxgd@mail.nwpu.edu.cn (H.L.)

**Keywords:** wireless sensor network, compressed sensing, measurement matrix, time division multiple access

## Abstract

Compressed sensing theory has been widely used for data aggregation in WSNs due to its capability of containing much information but with light load of transmission. However, there still exist some issues yet to be solved. For instance, the measurement matrix is complex to construct, and it is difficult to implement in hardware and not suitable for WSNs with limited node energy. To solve this problem, a random measurement matrix construction method based on Time Division Multiple Access (TDMA) is proposed based on the sparse random measurement matrix combined with the data transmission method of the TDMA of nodes in the cluster. The reconstruction performance of the number of non-zero elements per column in this matrix construction method for different signals was compared and analyzed through extensive experiments. It is demonstrated that the proposed matrix can not only accurately reconstruct the original signal, but also reduce the construction complexity from O(MN) to $O(d^{2}N)$ (d≪M), on the premise of achieving the same reconstruction effect as that of the sparse random measurement matrix. Moreover, the matrix construction method is further optimized by utilizing the correlation theory of nested matrices. A TDMA-based semi-random and semi-deterministic measurement matrix construction method is also proposed, which significantly reduces the construction complexity of the measurement matrix from $O(d^{2}N)$ to O(dN), and improves the construction efficiency of the measurement matrix. The findings in this work allow more flexible and efficient compressed sensing for data aggregation in WSNs.

## 1. Introduction

The wireless sensor network (WSN) is a distributed sensing network containing a certain number of endpoints (i.e., various sensors), which can be used to sense and detect the outside world [1]. Since WSNs can obtain massive amounts of objective physical information, they have been widely applied in different areas, including military defense, industrial and agricultural control, urban management, biological medical, environmental monitoring, rescue and disaster relief, and remote control of hazardous areas [2]. Due to the wireless nature of the sensors, they do not have a continuous supply of energy. As a result, the shortage and charging of energy has become a vital problem in the context of the sensor nodes. Data aggregation is an ideal and appropriate technology to address the energy consumption problem of the nodes [3,4,5,6,7]. By utilizing data aggregation technology, the information obtained from different nodes can be aggregated directly, which can significantly save the nodes’ energy.

In particular, the compressed sensing (CS) theory [8] is widely used for WSN data aggregation. The CS method can obtain a large amount of original information at the cost of transmitting a rather small amount of data, which is in accordance with the WSN data aggregation technology. It is worth stressing that the data collected by the WSN should be time and space dependent to meet the requirements of CS theory for compressible signals. In addition, CS theory is simple at the encoding side while being complex at the decoding side. This means it is well suited to the characteristics of WSNs with limited energy consumption at the front-end collection nodes and powerful aggregation nodes at the back end. Owning to the particular advantages of CS theory, data aggregation techniques for WSNs based on CS have become a promising architecture [8] since it was first proposed by Bajma et al. [9].

The construction of the measurement matrix is an essential prerequisite to utilize CS theory. In WSNs, it is desirable to obtain the highest possible accuracy in data reconstruction at the cost of the least data used. This strongly depends on the measurement matrix, which both affects the reconstruction accuracy of the signal and determines the amount of data to be transmitted. Therefore, since the emergence of CS theory, an increasing number of methods for constructing measurement matrices have been proposed. They can be broadly classified into two main categories: the random measurement matrices and the deterministic measurement matrices.

Random measurement matrices have randomly generated elements, such as Gaussian random matrices [10], Bernoulli random matrices [11], Fourier random matrices [12], partially orthogonal matrices [13], and sparse random matrices [14]. It has been shown that these random matrices have good independence and high probability in precisely reconstructing the original data. However, the hardware implementation of the matrices is complicated due to their random nature. To account for this issue, deterministic measurement matrices were proposed, which include edge-inflated graph adjacency matrices, polynomial deterministic matrices, coding matrices, chaotic sequences [15], Chirp codes [16], Toplitz rotation matrices, chunking matrices, and nesting matrices. Compared with random measurement matrices, deterministic measurement matrices are relatively easy to implement in terms of hardware. Nevertheless, the vast majority of deterministic measurement matrices are dense matrices (i.e., not sparse), meaning they are not suitable for WSNs where node energy is limited.

In order to reduce the complexity of the measurement matrix construction, and simplify and facilitate the implementation in hardware, some proper matrices have been constructed, e.g., the Vandermonde matrix [17], the random spacing sparse matrix [18], and the sparsest random block diagonal measurement matrix [19]. A unitary matrix and a sparse random matrix have been combined to propose a new dual-structured measurement matrix [20]. However, the matrix only works effectively when N<2M, which limits the applied range of the matrix. A novel compressive sensing method based on a singular value decomposition sparse random measurement matrix is proposed in [21]. Compared with the Toeplitz matrix, it requires a smaller number of independent random variables. However, this matrix is still a dense matrix, which requires a large amount of data to be transmitted and is not suitable for WSNs with limited node energy. Determining how to construct a measurement matrix with high reconstruction accuracy, small data transfer volume, and easy implementation is still a problem that needs to be urgently solved.

In this study, it was experimentally found that the use of sparse random matrices in WSNs can be further optimized. Based on these findings, a novel method used to construct the measurement matrices is proposed. The proposed method is an optimization of the sparse random matrix, which reduces the complexity of matrix construction and makes it easier to implement in hardware. Simulation experimental results show that the proposed method can achieve a reconstruction performance consistent with that of sparse random matrices, thus verifying the correctness and applicability of this method. The key idea of this method is to fix the number of non-zero elements in each column of the matrix and to randomly generate only the positions of the non-zero elements so as to reduce the complexity of the matrix construction.

The remainder of this paper is organized as follows: Section 2 compares the performance of four commonly used measurement matrices in terms of data reconstruction accuracy. Section 3 presents a random measurement matrix construction method based on TDMA in conjunction with sparse random matrices, and further optimizes the construction method to obtain a semi-random semi-deterministic measurement matrix construction method based on TDMA. In Section 4, the performance of the proposed method is verified by the simulation experiments. Finally, conclusions are presented in Section 5.

## 2. Common Measurement Matrix and Performance Verification

### 2.1. CS Theory Overview

According to CS theory, the sparse or compressible signals [22] can be sampled at a much lower frequency than the Nyquist sampling frequency, and can be perfectly reconstructed by a nonlinear reconstruction algorithm. That is, for an *N*-dimensional sparse signal **x**, it can be sparsely decomposed under a *M* × *N*-dimensional sparse transformation matrix **Ψ** as:(1)x=Ψθ,
where **θ** is an *k*-sparse *N*-dimensional column vector, that is, there are only *k* non-zero terms in **θ** and *k* is much smaller than *N*. Then, projecting under the *M* × *N*-dimensional measurement matrix **Φ**, an *M*-dimensional observation **y** can be obtained:(2)y=Φx=ΦΨθ=Tθ,
where *M* is much smaller than *N*, and **T** is called the sensing matrix.

Candès et al. [23] gave a sufficient condition for the existence of a deterministic solution to Equation (2) which requires that **T** satisfies the Restricted Isometry Property (RIP). Specifically, a matrix **T** is said to satisfy RIP if there exists *δ* ∈ (0, 1) such that every *k*-sparse signal **θ** satisfies Equation (3):(3)$$(1-\delta )\|\theta {\|}_{2}^{2}\leq \|T\theta {\|}_{2}^{2}\leq (1+\delta )\|\theta {\|}_{2}^{2},$$

Thus, the reconstructed signal can be obtained by minimizing the *l*_0_ norm or *l*_1_ norm of **θ** as:(4)$$\theta^{\prime }=\arg\min\|\theta {\|}_{0}\;\mathrm{or}\;\theta^{\prime }=\arg\min\|\theta {\|}_{1},$$

The above problem can be approximated utilizing linear programming or other convex optimization algorithms [24], which enables accurate reconstruction of the *k*-sparse signal **θ** with high probability.

Although the RIP theory is widely applied in constructing measurement matrices, proof of whether the matrices satisfy RIP appears to be very difficult to obtain under the theoretical framework of RIP. To simplify the derivation of RIP, Donoho et al. [22] optimized the measurement matrix design using the column coherence from the exact reconstruction condition of the compressed sampled signal. Tropp et al. [24] pointed out that the pursuit algorithms [e.g., basis pursuit (BP) and orthogonal matching pursuit (OMP)] can achieve accurate sparse approximation of the original signal when the redundant dictionary satisfies specific column coherence conditions. On this basis, Li et al. [25] proposed the column coherence-based theory for the measurement matrix construction.

For the sake of completeness, a brief introduction to the column coherence method is given as follows. The interdependence coefficient *μ* for an arbitrary matrix **U** is first defined as:(5)$$\mu =\max\left|\langle U_{i},U_{j}\rangle \right|,i\neq j,$$

For any *M* × *N*-dimensional real matrix with *M* being much smaller than *N*, the lower bound is given as:(6)$$\mu \geq \sqrt{\frac{N-M}{M(N-1)}}$$.

According to the inaccuracy principle [26], it can be obtained that the *l*_1_ and the *l*_0_ regularization problems are equivalent (i.e., the solution is unique) when the sparsity of the sparse signal **θ** satisfies:(7)$$\|\theta {\|}_{0}\leq \frac{1}{2}(1+\frac{1}{\mu \{T\}}),$$
where μ{T} is the interdependence coefficient of the sensing matrix **T**.

In order to satisfy (7), it is required to design the measurement matrix **T** with the smallest interdependence coefficient since the sparsity, which is the left-hand side of (7), is determined only by the signal itself. Compared with RIP, the column coherence condition is much simpler and more efficient. It can be proved that the perceptual factor satisfies the RIP in the case of satisfying the column coherence condition.

### 2.2. Common Measurement Matrices

#### 2.2.1. Gaussian Random Measurement Matrix

The Gaussian random measurement matrix is the most widely applied method in CS. For a Gaussian random measurement matrix **Φ** with the dimension of *M* × *N*, each element of **Φ** obeys the Gaussian distribution with zero mean and variance 1/*M* independently:(8)$$\Phi_{ij}∼N(0,\frac{1}{\sqrt{M}}),$$

The Gaussian random measurement matrix is totally random. It can be obtained that the RIP conditions can be satisfied with high probability when the number of measurements of the Gaussian random measurement matrix satisfies:(9)$$M\geq ck\log(\frac{N}{k}),$$
where *c* is a small constant and *k* is the sparsity of the measured signal. The Gaussian random matrix is uncorrelated with most of the sparse bases and can therefore be used as a universal measurement matrix.

#### 2.2.2. Bernoulli Random Measurement Matrix

The Bernoulli random measurement matrix has similar properties as the Gaussian random measurement matrix. For a Bernoulli random measurement matrix **Φ** with the dimension of *M* × *N*, each element of **Φ** obeys the Bernoulli distribution independently. Specifically, the element of the matrix can be represented as:(10)$$\Phi_{ij}=\left\{ \begin{array}{ll} \frac{1}{\sqrt{M}}, & P=\frac{1}{2}\\ -\frac{1}{\sqrt{M}}, & P=\frac{1}{2} \end{array}\right.,$$

As with the Gaussian random measurement matrix, the RIP condition is likely to be satisfied when the number of measurements of the Bernoulli random measurement matrix satisfies (9). Each element of this matrix has only two values, so it is easier to construct than the Gaussian random measurement matrix.

#### 2.2.3. Sparse Random Measurement Matrix

Both Gaussian and Bernoulli random matrices are dense matrices; however, Zhao H et al. [27] pointed out that a smaller number of sparse projection values possess most of the information of the original signal, based on which a sparse random measurement matrix was proposed. The element of the sparse random matrix is defined as:(11)$$\Phi_{ij}=\sqrt{\alpha }\left\{ \begin{array}{ll} 1, & P=\frac{1}{2\alpha }\\ 0, & P=1-\frac{1}{\alpha }\\ -1, & P=\frac{1}{2\alpha } \end{array}\right.,$$
where α is a constant that determines the sparsity of the matrix. Obviously, as α increases, the number of 0 elements in **Φ** rises, and the degree of sparsity is enhanced. In practice, it is shown that this matrix has a high accuracy of data reconstruction in the following cases:(12)$$\alpha =\sqrt{N},$$

#### 2.2.4. Toeplitz and Circulant Measurement Matrix

The general Toeplitz and circulant matrices [28] have the following form:(13)$$T=\left[\begin{array}{cccc} t_{n} & t_{n-1} & \cdots & t_{1}\\ t_{n+1} & t_{n} & \cdots & t_{2}\\ \vdots & \vdots & & \vdots \\ t_{2n-1} & t_{2n-2} & \cdots & t_{n} \end{array}\right]\;\;\;C=\left[\begin{array}{cccc} t_{n} & t_{n-1} & \cdots & t_{1}\\ t_{1} & t_{n} & \cdots & t_{2}\\ \vdots & \vdots & & \vdots \\ t_{n-1} & t_{2n-2} & \cdots & t_{n} \end{array}\right],$$
where **T** is the Toeplitz matrix and **C** is the circulant matrix. The Toeplitz and circulant measurement matrix is generated by cyclic displacement of row vectors. The circular displacement is easy to implement in hardware, meaning that the general Toeplitz and circulant matrix is promising in related areas.

### 2.3. Reconstructing Performance Validation

Extensive simulations were designed to compare and analyze the performance of the common measurement matrices introduced above. The grayscale image file lena.bmp in Figure 1a, which is commonly used in signal processing, was selected as the original data.

The cosine sparse matrix was used as the sparse basis, and the grayscale image formed by the sparse representation of this original image has a high sparsity under the cosine sparse representation, which satisfies the conditions for the use of CS theory. The four matrices mentioned above were used as measurement matrices, and data reconstruction was performed using the OMP algorithm.

Data reconstruction accuracy can be evaluated by the mean absolute error (MAE) [29]:(14)$$MAE=\frac{{\left\|x-x_{r}\right\|}_{2}}{{\left\|x\right\|}_{2}},$$
where x denotes the original data and $x_{r}$ denotes the reconstructed data. The data reconstruction accuracy of the four measurement matrices at different sampling frequencies is shown in Figure 2. It is clear that all four measurement matrices can ultimately reconstruct the original data accurately, and that the reconstruction error of all four measurement matrices decreases significantly when the sampling rate is greater than 0.2.

The column interdependence coefficients of the four measurement matrices as functions of the sampling rates are shown in Figure 3. It can be seen that the column interdependence coefficients of the Toeplitz and circular matrix are slightly higher than the other three matrices. It is well known that the interdependence coefficients of the measurement matrices are expected to be as small as possible. Therefore, it is natural for this matrix to have a higher reconstruction error, which is verified and exhibited in Figure 2. The reconstruction error of this matrix is slightly higher than the other three matrices until the sampling rate reaches 0.5.

The amount of data required to be transmitted by the four measurement matrices in reconstructing the data as a function of the sampling rates is shown in Figure 4. It can be seen that the amount of data required to be transferred by the sparse random matrix is the smallest; this is because the sparse random matrix is the sparsest.

Combining the analyses of the simulation results above, it can be concluded that among these four matrices, the sparse random matrix can provide a comparable reconstruction accuracy to that of the other matrices at the cost of a lower amount of data. However, due to the completely random nature of the sparse random matrix, it is harder to implement and realize in hardware compared to Toeplitz and cyclic matrices. Nevertheless, the performance of the sparse random matrices in WSN data aggregation can still be improved.

## 3. Measurement Matrix Construction Method based on TDMA

### 3.1. Random Measurement Matrix Construction Method Based on TDMA

The sparse random measurement matrix exhibits good performance in terms of data reconstruction. Note that each element in this matrix is randomly valued according to a certain probability; therefore, the construction complexity of this matrix is O(*MN*), where *M* is the number of matrix rows and *N* is the number of matrix columns. Thus, it is difficult to implement on sensor nodes with limited energy. To make it easier to implement in hardware, a TDMA-based method for constructing a random measurement matrix is proposed by modifying the sparse random measurement matrix. This is constructed as follows: first, several nodes generate projection vectors within their own TDMA time slots, and then the measurement matrix corresponding to these nodes can be constructed using the projection vectors as column vectors.

Figure 5 shows a schematic diagram of TDMA time slots for nodes in a cluster, where the number indicates the time slot assigned to the node. It is assumed that the data collected by sensor node *i* in a time slot is ${\left[x_{1},x_{2},\cdots x_{n}\right]}^{T}$. They have a strong temporal correlation since they are collected by one node in one time slot. Then, the data collected by the nodes in the cluster shown in Figure 5 can be constructed in the form of the following matrix:(15)$$x=\left[\begin{array}{cccc} x_{11} & x_{12} & \cdots & x_{19}\\ x_{21} & x_{22} & \cdots & x_{29}\\ \vdots & \vdots & & \vdots \\ x_{n1} & x_{n2} & \cdots & x_{n9} \end{array}\right],$$
where the data collected by each node within a time slot constitutes the column vector of the matrix. There is a strong temporal correlation between single columns of data in this matrix, and a strong spatial correlation between the different columns since they are all nodes within a cluster. To take full advantage of the temporal and spatial correlation between the data collected by this network of nodes, **x** is reshaped into a column vector when using the CS data aggregation; then, **y** can be represented as:(16)$$y=\left[\begin{array}{cccccc} \phi_{11} & \phi_{12} & \cdots & \phi_{1n} & \cdots & \phi_{1N}\\ \phi_{21} & \phi_{22} & \cdots & \phi_{2n} & \cdots & \phi_{2N}\\ \vdots & \vdots & & \vdots & & \vdots \\ \phi_{M1} & \phi_{M2} & \cdots & \phi_{Mn} & \cdots & \phi_{MN} \end{array}\right]\times \left[\begin{array}{c} x_{11}\\ x_{21}\\ \vdots \\ x_{n1}\\ \vdots \\ x_{19}\\ x_{29}\\ \vdots \\ x_{n9} \end{array}\right].$$

As shown in Equation (16), the corresponding encoding vector of the elements $x_{11}$ is ${\left[\begin{array}{cccc} \phi_{11} & \phi_{21} & \cdots & \phi_{M1} \end{array}\right]}^{T}$. This coded vector is constructed as follows:The node generates a list of *d* integers from [1,2,⋯M] with equal probability and no repetition to form the position list D. These *d* integers represent the positions of the *d* non-zero values in the vector.A node randomly sorts a list based on the principle that the number of +1 is d/2 and the number of −1 is also d/2 to obtain a list of symbols I.The element at this coded vector D[i] is I[i], where i=1,2,⋯d, and the elements at the remaining positions are all 0.

The elements of the constructed measurement matrix are then shown as:(17)$$\Phi_{ij}=\{\begin{array}{ll} I_{j}[k], & i=D_{j}[k],k=1,2,\ldots d\\ 0, & \mathrm{others} \end{array},$$
where $\Phi_{ij}$ denotes the element of the *i*-th row and *j*-th column of the measurement matrix Φ, $I_{j}[k]$ indicates whether the *k*-th non-zero element of the *j*-th column is positive or negative, and $D_{j}[k]$ represents the position of the *k*-th non-zero element of the *j*-th column.

Expressed in terms of probabilities, Equation (17) can be represented as:(18)$$\Phi_{ij}=\left\{ \begin{array}{ll} 1, & P=\frac{d}{2M}\\ 0, & P=1-\frac{d}{M}\\ -1, & P=\frac{d}{2M} \end{array}\right.,$$

Comparing Equations (11) and (18), if we denote
(19)$$\frac{1}{\alpha }=\frac{d}{M},$$
and multiply each element of the list I by $\sqrt{M/d}$, then we can obtain:(20)$$\Phi_{ij}=\sqrt{\frac{M}{d}}\left\{ \begin{array}{ll} 1, & P=\frac{d}{2M}\\ 0, & P=1-\frac{d}{M}\\ -1, & P=\frac{d}{2M} \end{array}\right.,$$

Obviously, comparing (20) and (11), the proposed matrix has the same probability representation as that of the sparse random matrix.

**Remark** **1.**
*It should be noted that the proposed method is quite different to the sparse random measurement matrix. Compared with the sparse random measurement matrix, which is generated obeying a certain distribution [i.e., Equation (11)] with a complexity of O(MN), the proposed TDMA-based random measurement matrix only needs to generate d random values with one random ordering, leading to a better construction complexity $O(d^{2}N)$ (where d≪M≪N). It can be seen that the complexity of the construction of the measurement matrix is greatly reduced.*


### 3.2. Semi-Random Semi-Deterministic Measurement Matrix Construction Method Based on TDMA

Aiming to further reduce the complexity of the measurement matrix implementation and simplify the implementation in hardware, a semi-random semi-deterministic measurement matrix construction method based on TDMA is proposed, which uses a nested matrix to nest the above random matrix with the deterministic matrix.

Ullah K et al. presented the following theorem on the correlation of matrices [30,31]. Assume that the number of non-zero elements in each column of the matrix $V_{M\times N}$ is *P* and the column coherence coefficient is μ(V). The matrix W is of size P×K with all elements having the same absolute values. Then it is possible to construct a nested matrix $Z_{M\times NK}$, and the column coherence coefficient of matrix Z satisfies the following conditions:(21)μ(Z)≤max(μ(V),μ(W)),

The construction process of the matrix Z is shown in Figure 6 [32,33]. The *i*-th non-zero element in matrix V is replaced by the *i*-th row in matrix W, and the zero elements are extended to the same dimension until all non-zero elements in the matrix are replaced.

According to the above theorem, the TDMA-based random measurement matrix construction method proposed in this paper can be optimized using the following procedure:The number of non-zero elements in each column of the measurement matrix Φ is *d*, which satisfies the requirement of the above theorem that the number of non-zero elements in each column of the matrix V is equal. So Φ is used as V in Equation (21).An orthogonal matrix of size d×d is used as W to generate the nested matrices, because the orthogonal matrix has the smallest column coherence coefficient. Furthermore, to ensure that the nested matrix satisfies Equation (20), the values of the elements in the matrix W are set to $\pm \sqrt{M/d}$, which also satisfies the requirement of the above theorem that all elements of W have the same absolute value.The final measurement matrix Φ is constructed in the manner of Figure 6.

**Remark** **2.**
*Compared with the measurement matrix constructed by the original method, the reconstructed effect of the semi-random and semi-deterministic matrix is essentially the same in theory as the original because its column coherence coefficient does not increase. In terms of construction complexity, the nested matrices generate d columns from one column of matrix Φ, so it only needs to construct N/d columns instead of N columns. Therefore, the construction complexity is reduced from $O(d^{2}N)$ to O(dN), making the hardware implementation simpler.*


## 4. Simulation Experiments and Result Analysis

### 4.1. Comparative Analysis of Different d Values on the Reconstruction Performance

In order to verify the influence of the value of the parameter *d* on the reconstructed performance of the measurement matrix, extensive simulations were performed. Specifically, different values of *d* in different signals were compared and analyzed. The original signal uses an artificially sparse random signal, which is itself sparse and does not need to be sparsely expressed. The original signal has length *N* and sparsity *k*. It takes on a random value from 1 to 100. The OMP algorithm is used to reconstruct the signal, and the MAE is adopted to evaluate the reconstruction accuracy.

#### 4.1.1. Simulation Results with the Length 100, Sparsity 10 and d 2, 4, 6, 8, 10

The original signal with length 100 and sparsity 10 is shown in Figure 7a. The measurement matrix is constructed by choosing 2, 4, 6, 8, and 10 as the *d* values. Because of a large number of random factors involved in this experimental procedure, the results of the experiment were averaged over 100 independent replications. The reconstructed accuracy of the measurement matrix for different values of *d* is shown in Figure 7b. From this figure, it can be seen that the proposed measurement matrix has a certain degree of universality. When the compression rate is greater than a certain value (in this case around 0.5), the original signal can be accurately reconstructed. In this experiment, the reconstruction accuracy of the measurement matrix with *d* = 2 is poor, whereas the reconstruction accuracy of the remaining *d* values is generally consistent. In addition, the MAE values greater than 1 appear at small sampling rates. This seems to not be normal but the reason is as follows: the original signal sizes generated randomly in this paper are all positive, but some elements of the reconstructed signal become negative when the sampling rate is too low, thus making the error greater than 1.

Furthermore, this experiment was repeated independently 1000 times to count the number of accurate reconstructions of the original signal for different values of d. The corresponding results are shown in Figure 8. The threshold value of MAE is set to 0.1. The experiment result less than the threshold is declared to be an accurate reconstruction of the original signal. Clearly, the reconstruction effect becomes worse as the sampling rate increases, and reaches the worst value when *d* = 2. Even with a sampling rate of 1, the original signal can only be accurately reconstructed 844 times. The reconstruction results for the other *d* values are essentially equivalent, and the original signal can be accurately reconstructed with a high probability at a sampling rate of 0.5.

#### 4.1.2. Simulation Results with the Length 200, Sparsity 5, 10, 15, 20 and d 2, 4, 6, 8, 10

The original signal length is taken as 200, the sparsity is taken as 5, 10, 15, and 20, and the value of *d* is taken as 2, 4, 6, 8, and 10. The experiments were independently repeated 100 times, and the threshold value of MAE was set to 0.1. The number of times the measurement matrix could accurately reconstruct the original signal with different *d* values and different signal sparsity is shown in Figure 9. It can be seen that when *d* = 2, the original signal can eventually be accurately reconstructed with high probability only at a sparsity of 5 as the sampling rate increases, and the original signal cannot be accurately reconstructed with high probability in all other sparsity cases.

For the remaining *d* values, the measurement matrix reconstruction performances are comparable, and all matrices can accurately reconstruct the original signal with high probability as the sampling rate increases. Furthermore, the sampling rate increases with the increasing sparsity (around 0.2 at k=5 and around 0.5 at k=20), so as to accurately reconstruct the original signal with high probability. An increase in sampling rate implies an increase in the number of rows *M* of the measurement matrix Φ. That is, an increase in *M* leads to a better reconstruction. This is probably because the proposed measurement matrix construction method only has *d* non-zero elements in each column, the positions of which are chosen randomly in *M*. Thus, an increase in *M* leads to an increase in the randomness of the *d* positions that can be selected, which in turn causes a decrease in the column correlation of the resulting measurement matrix. Therefore, higher probability of accurate reconstruction can be achieved.

#### 4.1.3. Simulation Results with the Length 500, 1000, 1500, Sparsity 60 and d 2, 4, 6, 8, 10

To highlight the fact that *d* ≪ *M* in the proposed method, the original signal length is set to 500, 1000, and 1500, the sparsity is set to 60, and the values of *d* are taken as 2, 4, 6, 8, and 10. The above simulation was repeated and the results are shown in the Figure 10. As can be seen from the figure, the sampling rate of the accurately reconstructed original signal is roughly around 0.6, 0.3, and 0.2, when *N* is 500, 1000, and 1500, respectively. The required number of samples *M* is about 250 for different signal lengths *N*. This indicates the signal length has basically no effect on the reconstruction performance for signals of the same sparsity. Furthermore, the value of M is taken to be proportional to the sparsity of the signal, which is approximately 4 times the sparsity of the signal. We can see that, in order to reconstruct the original data accurately, *M* is taken to be around 250, whereas at this point *d* is taken to be 4 or 6. This means that 250×N random values need to be created for the construction of the sparse random matrix, whereas only 6×N are needed for the proposed matrix, which greatly reduces the complexity of matrix construction.

### 4.2. Simulation Validation Based on Realistic Scenario

The source of experimental samples was sensors located in the laboratory [34]. Specifically, a total of approximately 2 million samples collected by 50 sensor nodes were utilized, which contain information on temperature, humidity, light, voltage, etc. For the sake of convenience and without loss of generality, the temperature samples over a certain period of time were used for the simulation. That is, 20 temperature samples collected by all the sensor nodes over the specific period (20×50=1000) were adopted as the original samples. The cosine sparse basis was used as the sparse matrix to reconstruct the data using the OMP algorithm. The sparse random matrix, the random measurement matrix based on TDMA, and the optimized semi-random semi-deterministic matrix were used as measurement matrices, respectively. For the sparse random matrix, the parameter α in Equation (12) was set to $\sqrt{N}$ to obtain a high reconstruction accuracy; for the random measurement matrix based on TDMA, the parameter *d* was set to 4; for the semi-random semi-deterministic matrix, W was chosen as follows:(22)$$W=\sqrt{\frac{M}{4}}\times \left[\begin{array}{cccc} 1 & 1 & -1 & 1\\ 1 & -1 & 1 & 1\\ -1 & 1 & 1 & 1\\ -1 & -1 & -1 & 1 \end{array}\right],$$
where *M* is the number of rows of the measurement matrix Φ.

The reconstruction performances of different measurement matrices were compared in terms of the reconstruction accuracy, the number of exact reconstructions in 100 experiments, the measurement matrix column coherence coefficient, and the amount of data required for reconstruction, as shown in Figure 11.

As can be seen from the figure, the two proposed measurement matrices have similar performances to that of the random sparse matrix in terms of reconstruction accuracy, number of reconstructions, and matrix column coherence coefficients. However, the matrix complexity of the proposed method is much lower than that of the random sparse matrix, as can be seen from Figure 11d.

This indicates that the proposed TDMA-based random measurement matrix and the TDMA-based semi-random semi-deterministic matrix have comparable reconstruction performance as that of the sparse random measurement matrix, but at the cost of much lower complexity. They require a constant amount of data, which does not increase with the sampling rate. This is a large advantage over the sparse random measurement matrix, which becomes more apparent as the sampling rate increases. Furthermore, the TDMA-based semi-random semi-deterministic matrix is the least complex and the easiest to construct in terms of implementation complexity.

## 5. Conclusions

This study considered the construction of the measurement matrix in data aggregation for WSNs based on CS. To reduce the complexity and improve the reconstruction accuracy, a TDMA-based random measurement matrix construction method is proposed on the basis of sparse random measurement matrix. The construction complexity is reduced from O(MN) to $O(d^{2}N)$ (d≪M) compared with the sparse random matrix. To further reduce the complexity, the method is optimized using a nested matrix, leading to a semi-random and semi-deterministic measurement matrix based on TDMA. The complexity is further reduced to O(dN). Finally, simulation experiments were performed to verify the measurement matrix construction complexity and reconstruction accuracy. Simulation results show that the two proposed measurement matrices have comparable reconstruction performance as that of the sparse random measurement matrix, but at the cost of much lower complexity. Moreover, as the degree of sparsity increases, the reduction in the matrix construction complexity of the proposed method becomes more obvious. Hence, greater superiority of the proposed method is observed.

It should be noted that communication loss occurs during the generation of measurement matrices. However, this does not influence the complexity of the measurement matrix construction, which is the focus of this paper, because the energy is lost regardless of whether the matrix is generated for transmission by the sensor nodes to the sink node, or by the sink node to the sensor nodes. However, we will consider this energy loss in future work.

## Figures and Tables

**Figure 1 entropy-24-00493-f001:**
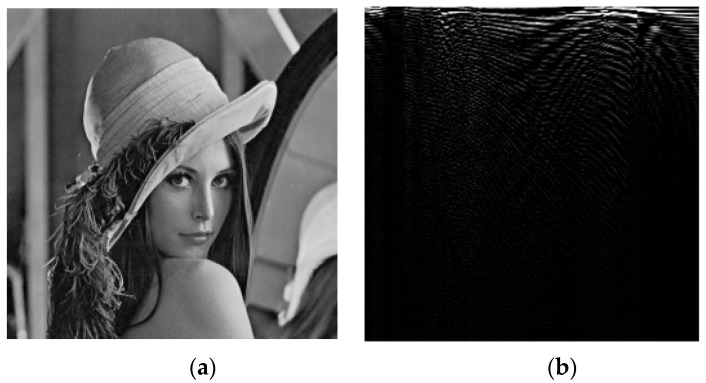
(**a**) Original picture; (**b**) the grayscale image formed by the sparse representation of this original image under the cosine sparse basis.

**Figure 2 entropy-24-00493-f002:**
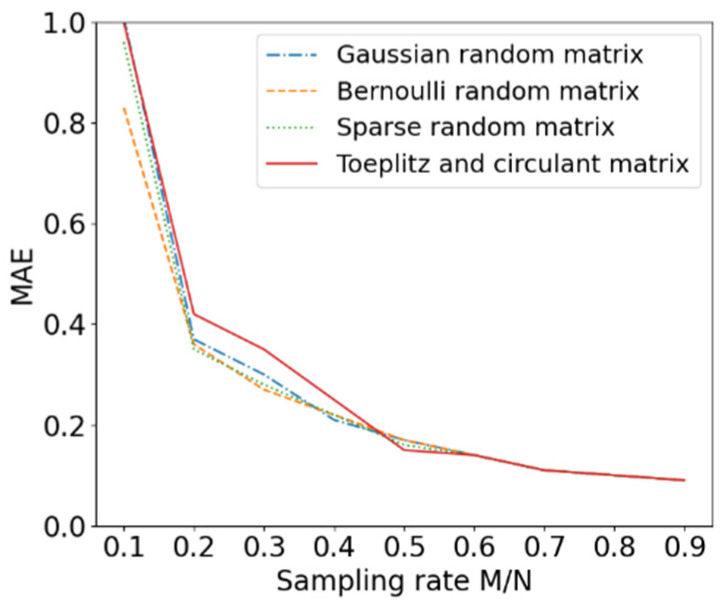
Reconstruction error comparison of four measurement matrices.

**Figure 3 entropy-24-00493-f003:**
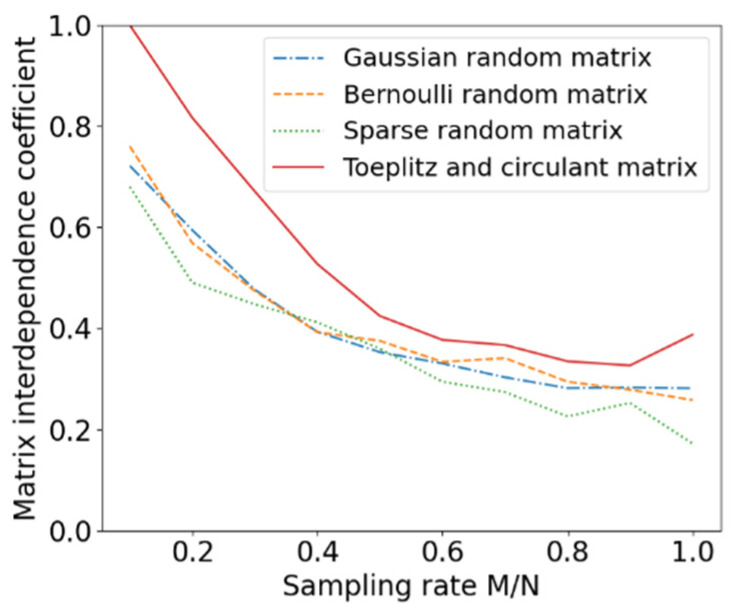
The column interdependence coefficients of the four measurement matrices.

**Figure 4 entropy-24-00493-f004:**
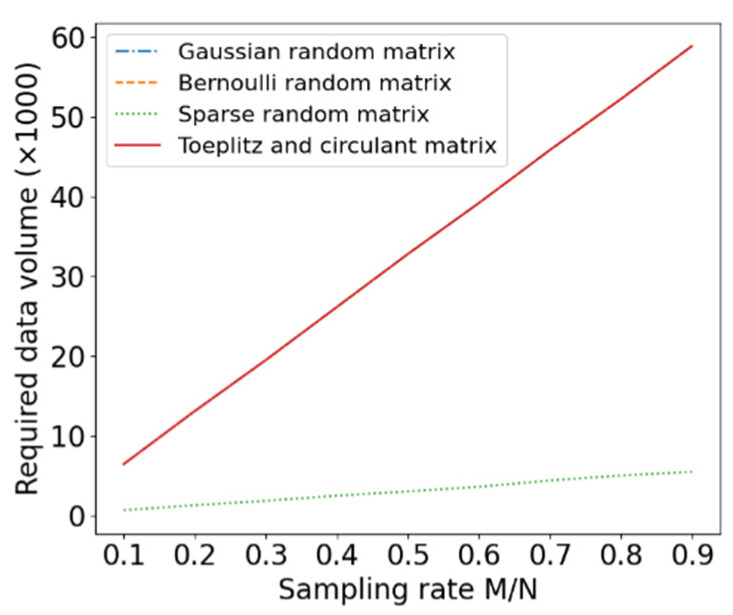
The amount of data to be transferred by the four measurement matrices.

**Figure 5 entropy-24-00493-f005:**
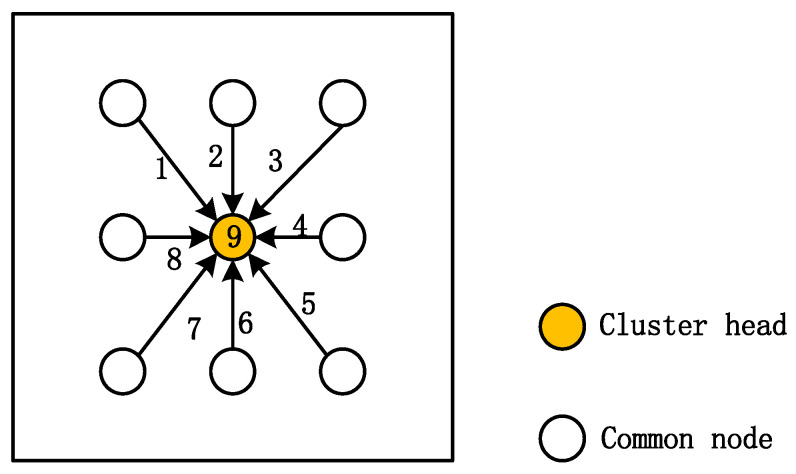
Diagram of TDMA time slot for nodes in a cluster.

**Figure 6 entropy-24-00493-f006:**
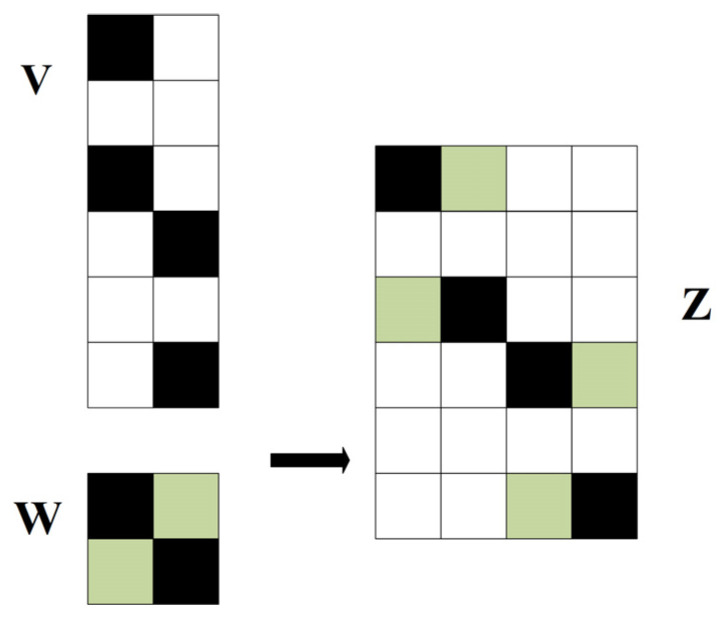
The construction process of nested matrices.

**Figure 7 entropy-24-00493-f007:**
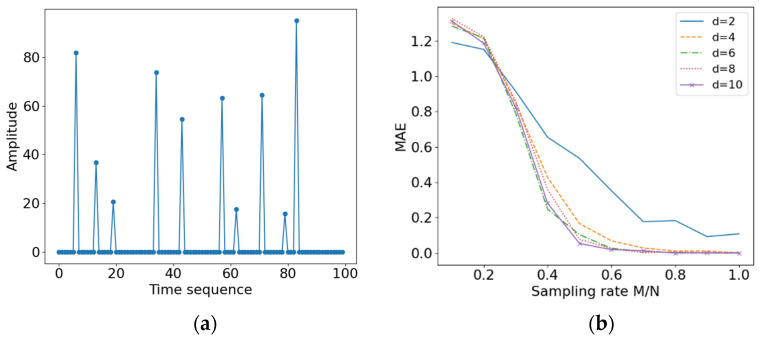
(**a**) Diagram of the original sparse random signal; (**b**) diagram of reconstruction accuracy of the TDMA-based random measurement matrix for different *d* values.

**Figure 8 entropy-24-00493-f008:**
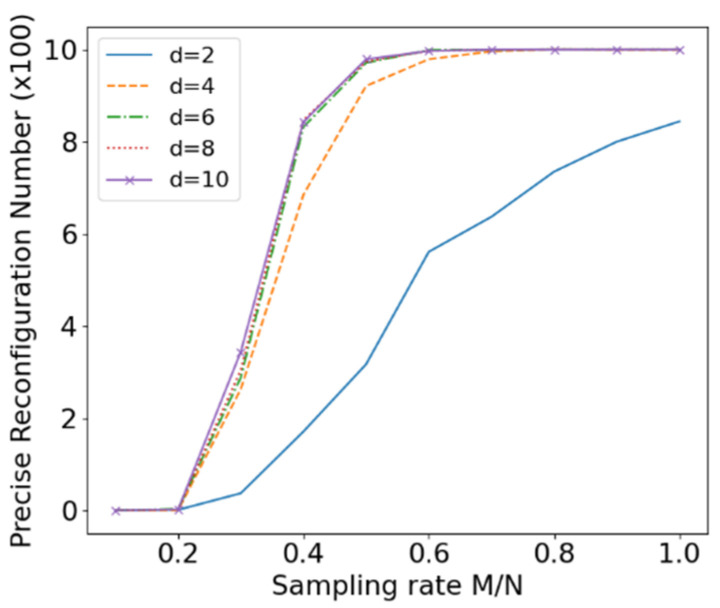
Number of times that the original signal can be accurately reconstructed.

**Figure 9 entropy-24-00493-f009:**
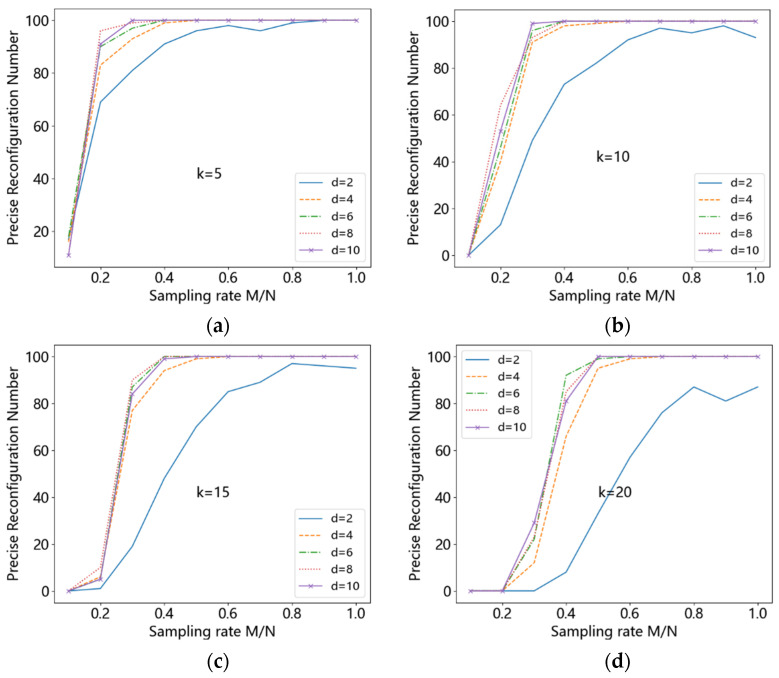
(**a**) k=5, (**b**) k=10, (**c**) k=15, (**d**) k=20.

**Figure 10 entropy-24-00493-f010:**
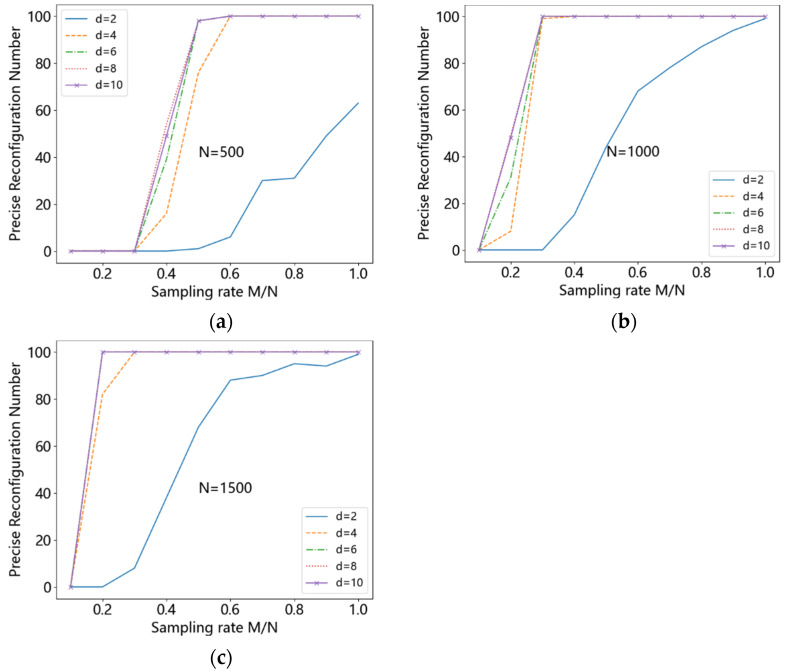
(**a**) N=500, (**b**) N=1000, (**c**) N=1500.

**Figure 11 entropy-24-00493-f011:**
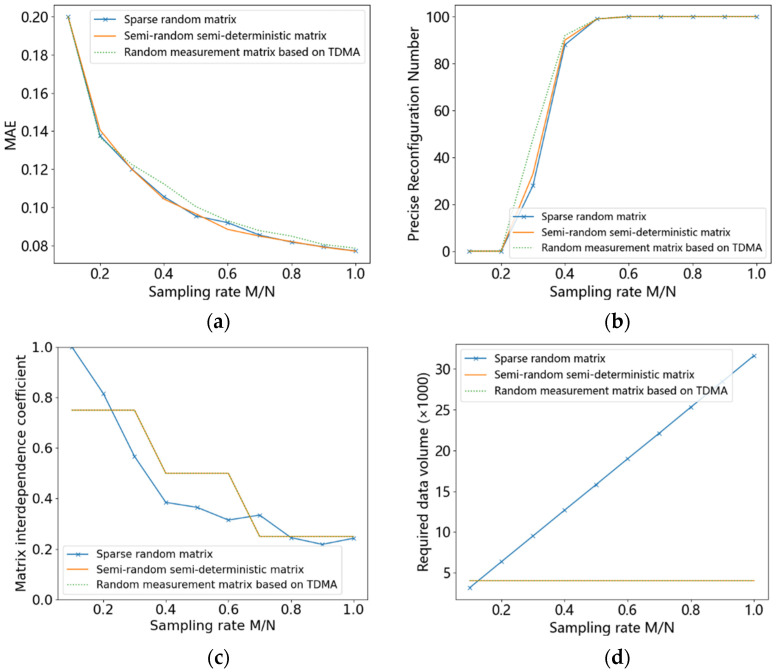
(**a**) The reconstruction accuracy, (**b**) the number of exact reconstructions in 100 experiments, (**c**) the measurement matrix column coherence coefficient, and (**d**) the amount of data required for reconstruction.

## Data Availability

The data sets for this study are available upon request from the corresponding author.

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
