# Peer review of "A Compressed Sensing Measurement Matrix Construction Method Based on TDMA for Wireless Sensor Networks"

_entropy, 2022, doi:10.3390/e24040493_

Round 1

Reviewer 1 Report

The authors propose a method for the construction of measurement matrices that optimises their use in WSN. 

-Please highlight the purpose of this study and provide a more complete description of the concept.

-In the introduction there is a lot of theory mentioned, however withour practical applications. 

-There is no reference of the dataset used, the link leads to all datasets in kaggle

-The paper doesn't have a conclusion section and ends abruptly. Which are the findings of the paper? There needs to be a discussion around the proposed method and the simulated results and also there needs to be a summary of the paper in the conclusion section also

-In the intro, the authors mention that there is a final conclusion section, which they probably forgot to add in the paper.

-The english needs editing in order to be more formal

-The references are limited, if this is because there is not much work in the field, it should have been stated clearly

Overall, the paper includes a detailed section of theory and mathematical formulas, however the scope of the paper is not visible, and the paper in general doesn't present a complete idea. Furthermore, the conclusion section is missing.

Reviewer 2 Report

This paper presents a CS-based compression method for wireless sensor networks. The authors suggest using TDMA to reduce the complexity of the random measurement matrix, but unfortunately, I cannot recommend the publication of this paper due to the following concerns.

  1. The authors claim that d << M, but it is not really true. M is basically (much) less than N. So, for example, in section 4.1.1., the value of M is at best on the order of tens, not significantly different from the value of d.
  2. As a result of the evaluation, the actual feasible d value seems to be 5 or more, which is not that small compared to M, making a limited gain.
  3. How can the MAE value be greater than 1 in Figure 7(b)?
  4. The authors do not consider the communication overhead for generating the matrix. 

Round 2

Reviewer 1 Report

The authors have made some additions in the paper, but it has not undergone major revision as suggested.  There are still some typos even in the newly added text. The authors need to put more effort in addressing the reviewers' comments  from the firts submission and be more careful with the structure and presentation of the paper. 

Author Response

We have made major revision as suggested this time. This time, we focused on the English expressions, structure and presentation, and conclusion of the paper. We have revised the English expressions throughout the paper to make it more professional. We have revised the structure and presentation of the paper to make it clearer. We have rewritten the conclusion of the paper to make it more understandable. You will see our efforts in our resubmitted manuscript, thanks for your reading!

Reviewer 2 Report

Much improvement have been made in this version, compared to the previous one, but the quality of figures, equations, notations, and presentations is not enough for  journal publication. For example, in equation (14), the authors are using the same notations for x. 

Author Response

First of all, thank you for your encouragementof us! We have improved the quality of figures, equations, notations and prensentations of this paper in order to it reaches the requirements of this journal. This time we have checked and revised the paper extensively and carefully. As for the occurrence of the same notations for x in equation (14), I need to explain it a little. We used  and  in the Word version, but when it was converted to the PDF version,  was shown as . We are not sure why this situation has happened, but of course we bear responsibility for not checking carefully. In the new manuscript, we have replaced  with . Thanks for your reading!

Round 3

Reviewer 1 Report

This is a much improved version of the paper. Just an advice for the future, when submitting the revised version it is better to show the newly added text, not the deleted text too, because it is difficult to read. The formatting changes don't need to be shown to the reviewers, you can accept only the formatting changes so it is an more appropriate reading presentation.

Reviewer 2 Report

Most of my previous concerns have been resolved.

This manuscript is a resubmission of an earlier submission. The following is a list of the peer review reports and author responses from that submission.

Round 1

Reviewer 1 Report

In this paper, the authors present a routing algorithm for wireless sensor networks, that may be helpful for reducing the total energy consumption. After reading it, my overall assessment is negative. See the follows:

1. The title is too much abstract; from the title, I thought that this paper would cover extensive WSN applications. It would be much better to highlight the proposed scheme in the title.

2. The introduction section is poorly written. It is very hard to understand what the authors are claiming and want to propose.

3. The authors must summarize the contributions of this paper in the introduction section for better understanding. Please refer to other papers. 

4. Related works should be added. To me it seems that the survey on previous approaches has not been sufficiently conducted. 

5. In section 2.2.1, the descriptions about matrices are poorly written. For example, for partial orthogonal measurement matrix, the authors just state that it it complicated so it's not applicable to WSN. How and why is it so complicated? 

6. There are too many mismatches between figure index and description.

7. The English language should be improved. 

Reviewer 2 Report

This paper is not well prepared and prepared. Numerous grammatical errors make the paper very hard to follow. What’s the contribution (if any) of the paper? How this experiment linked to wireless sensor networks?